# Same streams in a different forest? Investigations of forest harvest legacies and future trajectories across 30 years of stream habitat monitoring on the Tongass National Forest, Alaska

**Michael J. Moore J.**[1], **Rebecca L. Flitcroft**[2]*, **Emil Tucker**[3], **Katherine M. Prussian**[4], **Shannon M. Claeson**[5]

1 U.S. Geological Survey, Iowa Cooperative Fish and Wildlife Research Unit, Ames, IA, United States of America, 2 U.S. Forest Service, Pacific Northwest Research Station, Corvallis Forestry Sciences Laboratory, Corvallis, OR, United States of America, 3 U.S. Forest Service, Tongass National Forest, Petersburg, AK, United States of America, 4 U.S. Forest Service, Tongass National Forest, Sitka, AK, United States of America, 5 U.S. Forest Service, Pacific Northwest Research Station, Wenatchee Forestry Sciences Laboratory, Wenatchee, WA, United States of America

* rebecca.flitcroft@usda.gov

**Data Availability Statement:** The data underlying the results are being prepared and reviewed for

## Abstract

The effects of timber harvest practices and climate change have altered forest ecosystems in southeast Alaska. However, quantification of patterns and trends in stream habitats associated with these forests is limited owing to a paucity of data available in remote watersheds. Here, we analyzed a 30-year dataset from southeast Alaska's Tongass National Forest to understand how these factors shape stream habitats. First, we examined differences between broad management classes (i.e., harvested and non-harvested) that have been used to guide stream channel restoration goals. Second, we assessed associations between intrinsic landscape characteristics, watershed management, and timber harvest legacies on aquatic habitat metrics. And third, we examined trends in stream habitat metrics over the duration of the dataset to anticipate future management challenges for these systems. Small effect sizes for some harvest-related predictors suggest that some stream habitat metrics, such as pool densities, are less responsive than others, and management practices such as protecting riparian buffers as well as post-harvest restoration may help conserve fish habitats. Large wood densities increased with time since harvest at sites harvested >50 years ago, indicating that multiple decades of post-harvest forest regrowth may contribute large wood to streams (possibly alder), but that it is not enough time for old-growth trees (e.g., spruce, *Picea*, or hemlock, *Tsuga*,), classified as key wood, to develop and be delivered to streams. The declining trend in key wood (i.e., the largest size class of wood) regardless of management history may reflect that pre-harvest legacy old-growth trees are declining along streams, with low replacement. The introduction of wood to maintain complex stream habitats may fill this gap until riparian stands again contribute structural key wood to streams. Trend analyses indicate an increasing

release in the United States Forest Service's publicly available data repository and will be made available upon publication. In the meantime data and code are available from Rebecca Flitcroft, rebecca.flitcroft@usda.gov. 4 We have already discussed this with Publishing Editor, Emaan Basat who said "We can go forth with processing your manuscript without the data currently being available, I will simply add a note to say that the data will be available upon acceptance. Will the data be available upon acceptance? Also, the contact provided for the data (Rebecca Flitcroft) is a co-author on your manuscript, for data requests and questions we would require an ethics committee email contact or a non-author contact to prevent bias in accessing the data. Would you be able to provide this?" We have also discussed with Academic Editor, Alejandro Huertas-Herrera, as well via email.

**Funding:** No external funding was used in the analysis represented by this manuscript, rather the United States Forest Service providing salary to support the involvement of all authors.

**Competing interests:** The authors have declared that no competing interests exist.

spatial extent of undercut banks that may also be influenced by shifting hydrologic regimes under climate change.

## Introduction

The signature of human alteration to the climate and landscape is reflected on ecosystem condition on a global scale [1, 2]. Changes in the composition and structure of remote forested watersheds at northern latitudes are likely to accelerate, reflecting increasing global demand for natural resources, accumulation of persistent pollutants, and climate models that project atmospheric warming and increased frequency of droughts and floods. Southeast Alaska is characterized by forested systems distributed across numerous island and mainland areas. This area is the largest intact temperate rainforest on earth and includes the 68,800-km$^2$ (17-million-acre) Tongass National Forest (NF), the largest NF in the United States (Fig 1A) [3]. Management of NFs support a regional socio-economic system reliant on interconnected renewable natural resources such as fisheries, freshwater, and forestry [4]. From 2007 to 2019, the Tongass NF produced 75% of Southeast Alaska's commercial salmon harvest, with an average 2019 dockside annual value of US $68.5 million dollars per year [5]. Species such as coho salmon (*Oncorhynchus kisutch*), whose parr will remain in freshwaters for up to two years, and resident freshwater species such as Dolly Varden (*Salvelinus malma*) and cutthroat trout (*Oncorhyncus clarkii*) are sensitive to stream channel alteration owing to forestry practices, and thus are considered indicators of aquatic ecosystem health [6]. The forested landscape these streams flow through is changing as a result of ongoing forest management as well as climate change (i.e., temperature and precipitation).

Research on the responses of stream habitats to forest management in the coastal temperate forests of Alaska and Canada have focused on the effects of timber harvest on stream geomorphology and stream discharge [7–10], and the effects of harvest practices [10, 11], roads [12], large wood [13, 14], and sediment [15] on salmonid habitat. For example, logging roads can route overland runoff and fine sediments into stream channels, which may reduce salmon spawning-habitat suitability [12]. The supply of large wood pieces may be reduced post-harvest [16]. Wood is crucial for creating cover, capturing sediment, and forming low-velocity rearing pools for juvenile salmon and is dependent on riparian stand age and mortality rates [13, 14].

Teasing apart the effects of land-use legacies on habitat condition trajectories in a changing climate is challenging. Northern forests are predicted to change as a result of disturbances (e.g., invasive species, insects, disease, climate) that may alter tree growth, mortality, and phenology, with cascading consequences for terrestrial and aquatic biophysical systems [17, 18]. Climate change is predicted to affect river hydrologic regimes, which reflect patterns of precipitation and temperature. For instance, winter flow is lower in watersheds that receive more precipitation as snow as compared to rain-dominated systems, but snowmelt in these systems provides maintenance flows later into the summer than rain-dominated systems. Thus, as snow-dominated systems transition to rain-dominated systems, timing of water availability may also change, potentially disrupting phenological adaptation to environmental conditions for aquatic species and ecosystems [19]. Additionally, warmer temperatures may produce more intense winter rain in previously snow-dominated systems that may lead to flooding and streambed scour, altering channel morphology, and causing increased salmonid egg mortality [20, 21].

The impacts on Alaskan salmon populations from timber harvest, climate change, and stream restoration can vary spatially [22] and be species-specific owing to differences in life-

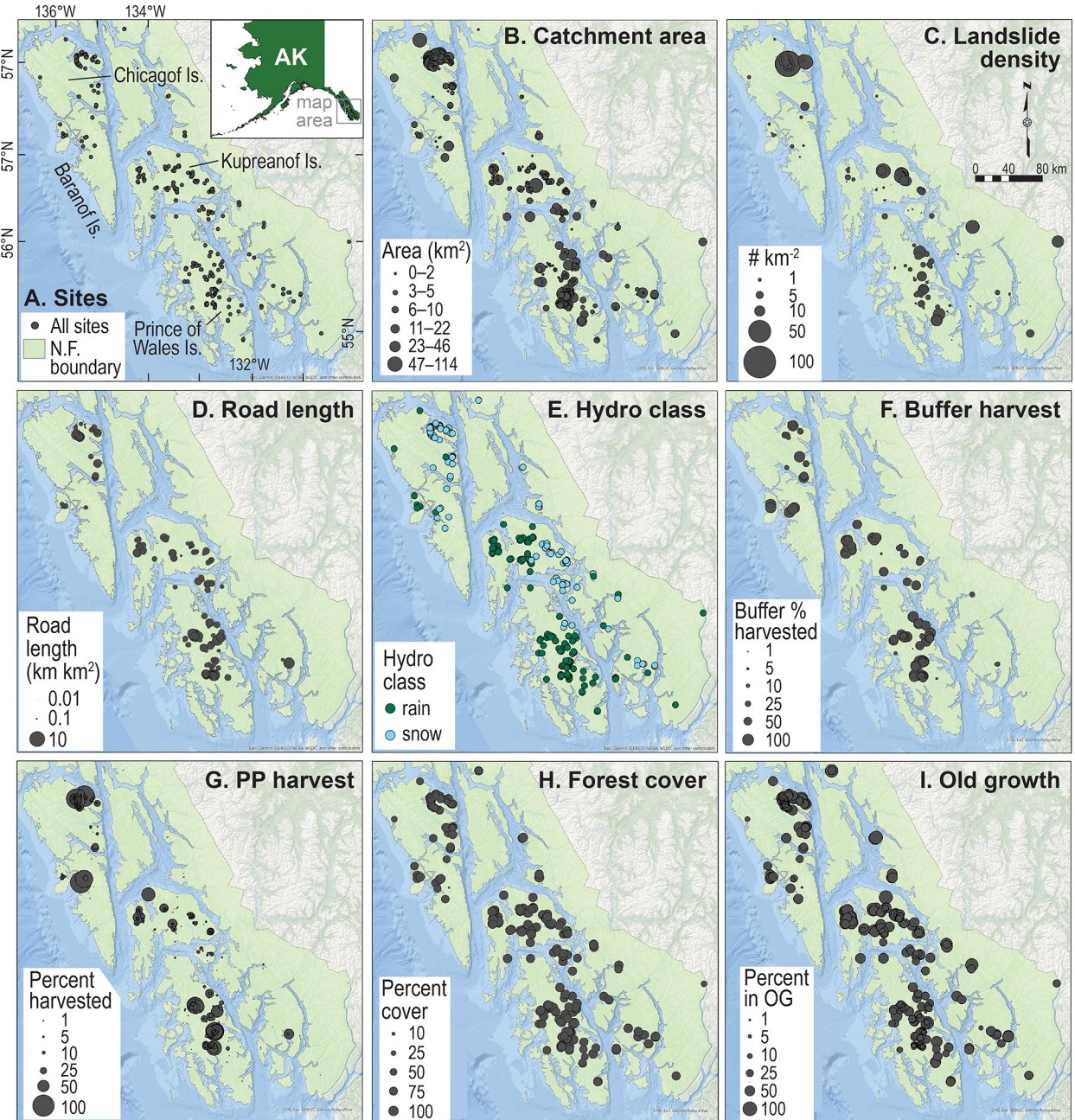

**Fig 1. Paneled study area map.** Panel A shows the location of each survey site and the location of the study area relative to the State of Alaska boundaries. The panels that follow depict the spatial distributions of 7 continuous variables and 1 categorical variable through the scaling of survey-site point size. Catchment area, also referred to as subwatershed area, is represented by a discrete graduated symbol-size scale. Continuous variables in other panels are represented by establishing a minimum point area for the smallest value of the variable and linearly increasing the point size proportional to that variable. Catchment area is the subwatershed drainage area (km²) using downstream site boundary as a pourpoint. Landslide density is the area of landslides in a catchment divided by catchment area. Road length is road length density calculated as length of roads in a catchment divided by catchment area. Hydroclass is a broad scale hydrologic classification for each survey site based on dominant precipitation type: rain (includes transitional rain–snow regimes) vs. snow. Buffer harvest is the proportion of the area of the stream analysis buffer that is 91 m on each side of study site that had been harvested prior to the site survey. PP (pourpoint) harvest is the proportion of a pourpoint catchment that was harvested prior to the site survey. Forest cover is the proportion of a catchment that was forested. Old growth is the proportion of the subwatershed area in size class 4 forest (>22.9 cm [>9 in] diameter at breast height or 137 cm [4.5 ft] above the ground) and defined as "old-growth sawtimber" >150 years old. Tongass National Forest (NF) boundaries are shaded green. Alaska state boundaries shapefile produced by the United States Census Bureau was accessed from: https://www.census.gov/geographies/mapping-files/time-series/geo/carto-boundary-file.html. ESRI World Oceans Basemap accessed from: https://www.arcgis.com/home/item.html?id=67ab7f7c535c4687b6518e6d2343e8a2; Sources: Esri, GEBCO, NOAA, National Geographic, DeLorme, HERE, Geonames.org, and other contributors.

history traits and habitat requirements [23]. Therefore, coordinated monitoring efforts to support science-driven management goals and actions may be valuable in detecting complex effects. Assessment methods for evaluating aquatic habitat response to forest-harvest practices can be divided into three designs: 1) long-term studies of experimental manipulations of a single site; 2) control and impact monitoring studies; and 3) extensive survey programs of sites with diverse management histories that substitute spatial replication for temporal replication [24]. Historically, uniform timber-harvest practices, implemented under what was considered to be a stable climate, allowed the Tongass National Forest (NF) to monitor streams with standard, broad management classification frameworks that contrasted stream habitats in watersheds that had any level of harvest versus reference watersheds that had none. This aligns more with the control and impact paradigm. For example, Bryant et al. [7] noted significant differences between sites in harvested and non-harvested watersheds in terms of total wood and pool densities, residual pool depth, and median particle size in some geomorphic stream classes.

However, multiple harvests and changing practices over many decades have produced a complex mosaic of forest stand ages on the landscape. Intensive early harvest practices clearcut riparian zones and, in some cases, removed in-channel wood. In contrast, more recent (since the mid-1990s) harvest events are smaller in scale and have included riparian buffer protections [3]. This variety of historical harvest practices and time since harvest among sites calls for a new assessment and monitoring approach. Assessment method 3 leverages newly available temporally and spatially expansive datasets collected using standardized habitat-assessment protocols. Under this assessment method, analysis of stream habitats on the Tongass NF would include statistical tools that control for variation to better explore the complex interacting consequences of changing timber-harvest management approaches, stream restoration, and climate change.

To facilitate ongoing adaptive management on the Tongass NF, we assessed stream habitat characteristics along a gradient of intrinsic characteristics and forest harvest practices using the most comprehensive long-term dataset assembled for southeast Alaska to-date. Analysis was completed following three sequential objectives. The first objective was to compare stream habitat conditions in harvested and unharvested watersheds on the Tongass NF using the historically applied control and impact classification framework. The second objective was to explore how gradients in landscape characteristics, legacies of forest management, and climate-driven hydrology may explain variation in stream habitat metrics on the Tongass using a more nuanced modeling approach to control for variation and disturbance magnitude. Finally, our third objective was to explore temporal trends in stream habitat metrics across the 30-year survey period to identify emerging management challenges. This work serves as a pathway to begin thinking of stream habitat characteristics in the Tongass along spatial and temporal harvest gradients similar to analyses that have been recently completed in other forested ecosystems of the Western United States [25–27]. We hypothesize that habitat differences between sites with harvested and unharvested watersheds will have decreased over time owing to declining harvest intensity, forest regeneration, and protection of riparian buffers. We expect that the effects of specific harvest-related, climate, and restoration variables such as time since harvest, hydrologic regime, and active wood placement in streams may be better identified through the mixed-model regression approaches in objectives 2 and 3. The purpose of objectives 2 and 3 is to explore correlative patterns in stream habitat conditions across space and time that may or may not match up with our expectations based on published literature. These patterns can serve as hypotheses to be examined with targeted, controlled research experiments in the future.

## Materials and methods

### Study site description

The study area has a coastal maritime climate characterized by cool temperatures and abundant rainfall ($> 200$ cm y$^{-1}$), which ranges widely owing to marine influence and topography. Substantial snow occurs at higher elevations on islands of the Alexander Archipelago or on the mainland during the winter. Thus, narrow ranges in winter temperatures have substantial effects on stream hydrologic regimes [28]. The Tongass NF was established in 1907. The Tongass Timber Act of 1947 opened harvest for old-growth Sitka spruce (*Picea sitchensis*), western hemlock (*Tsuga heterophylla*), western redcedar (*Thuja plicata*), and Alaska cedar (*Xanthocyparis nootkatensis*). The first timber harvests following the Tongass Timber Act occurred between 1950 and 1969, and were concentrated in easily accessed, productive, lowland watersheds near the coast [6]. In addition to the aforementioned species, red alder (*Alnus rubra*) is prevalent in open areas disturbed by logging, floods, and landslides [29]. Regulations to protect stream habitats during logging were instituted in the Tongass Timber Reform Act of 1990, which mandated a 33-m-wide protected "riparian management buffer" (RMB) along fish-producing streams in harvested tracts [30]. Data collection for this study began shortly after the institution of this Act and continued until 2020. For our analyses, we also defined a "stream analysis buffer" (SAB) using GIS, extending out 91 m (250 ft) on either side of a surveyed site. This stream buffer is wider than the Tongass NF regulated buffer but is similar to buffers afforded to streams on federal lands in other areas of the Pacific Northwest that are occupied by anadromous salmonids listed as threatened or endangered under the U.S. Federal Endangered Species Act [31]. Thus, the SAB is approximately 2.5 times the average tree height for 100-year-old trees [32] and is greater than the 33-m (100-ft) legally defined RMB width.

### Stream habitat metrics relevant for sensitive aquatic biota

We used a dataset consisting of 852 habitat surveys from 1991 to 2020 at 323 fish-bearing stream reaches (hereafter, sites) (Fig 1A) that were predominantly accessible by road (some sites required helicopter or off-road access). Sixty-three percent of the sites were surveyed more than once, with the number of repeat surveys ranging from 1 to 13, with a mean of between 2 and 3 surveys/site. We delineated the upstream drainage subwatersheds using a digital elevation model with survey sites corresponding to the downstream pourpoints. The subwatersheds were nested within 98 HUC-12 level watersheds (hydrologic unit codes, U.S. Geological Survey, National Hydrography Dataset https://www.usgs.gov/core-science-systems/ngp/national-hydrography).

The long-term habitat assessment database developed for this study comprises four major data-collection initiatives conducted as part of the Tongass National Forest Land and Resource Management Plan [33]; the Channel Condition Assessment (1996–1999, n = 28) [34]; general aquatic surveys (collected by the U.S. Forest Service (USFS); 1991–2017, n = 91); Management Indicator Species monitoring studies[6, 35], and the Watershed Restoration Effectiveness Monitoring study (collected by USFS, 2013–2020, n = 146). This compilation of datasets resulted in a non-uniform temporal and spatial distribution across the study area over time. For example, 90% of the surveys were conducted after 2000 and 30% between 2012 and 2015.

We completed a preliminary assessment of survey data to ensure consistent survey protocols. Most surveys measured habitat metrics that are invariable with discharge, correlate with fish populations, and are commonly measured in traditional habitat surveys to assess physical stream characteristics over space and time (e.g., Tier II surveys) (S1 Table in S1 File) [36]. Not

all metrics were measured in each stream survey, which depended on project-specific objectives. Sample sizes for individual metrics ranged from 453–784 surveys. The metrics analyzed in our multivariate and univariate statistical models and tests were associated with channel morphology (undercut bank density, width-to-depth ratio i.e., width:depth, pool density, pool spacing, residual pool depth, pool size, riffle area); large wood (large wood density and key wood density); or substrate transport (50th percentile particle size ($D_{50}$) and relative submergence) (S1 Table in S1 File).

## Categorical land management classification

Stream habitat assessments on the Tongass NF have been classified in the past based on harvest and land management history (i.e., no timber harvest versus harvest with or without riparian protections) as a simple way to facilitate stream restoration and to identify habitat metrics most influenced by timber harvest [7]. Therefore, we evaluated whether habitat differences found in previous analyses existed for the following three broad classes: 1) "reference" sites were identified as subwatersheds with <6% harvest area and no harvest within the SAB; 2) "harvested" sites had >6% subwatershed harvest area or some amount of SAB harvest; and "restored" sites which had past harvest but also had experienced restoration primarily through the installation of large or key wood pieces (see S2 Table in S1 File for description of key wood size thresholds) in the stream prior to site surveys.

## Landscape template as habitat predictors

Controlling for the natural variability in landscape characteristics allows us to better isolate management and climate effects on channel morphology and aquatic habitat [34]. Therefore, we compiled data from several GIS layers that represent intrinsic physiographic characteristics of subwatersheds (Table 1).

Channel gradient and confinement were accounted for with a categorical variable based on region-specific channel process groups that are used extensively by land managers on the Tongass NF. Owing to the limited sample size (see S3 Table in S1 File for model-specific sample sizes), we combined some of the process groups defined in Paustian et al. [37] based on gradient (S4 Table in S1 File). For example, all moderate-gradient process groups were combined into a single group. Final process groups used in our dataset include alluvial fan, floodplain, and low-, moderate-, and high-gradient reaches. We focus primarily on floodplain and moderate-gradient process groups because they are responsive to disturbances, generally contain productive forests that were targeted for historic harvest, and reflect the combined influences of habitat changes throughout upstream reaches of the stream network (S4 Table in S1 File) [34]. Floodplain systems are low-gradient alluvial depositional channels, situated in valley bottoms and lowlands with high stream flows not commonly contained within the active channel banks and where some degree of flood plain development is evident. The channels are predominantly composed of series of pools and riffles, with large wood the predominant pool forming mechanism. They often have more multi-threaded channels, greater sinuosity, and greater amounts of off channel habitats such as beaver ponds and sloughs [33]. The moderate gradient channels usually have channel gradients of 2–6 percent and occur near the transition between headwater streams and floodplain and alluvial fan channels. These streams usually have more confined valleys and coarser alluvial substrates ranging from gravel to boulder size comprising the channel beds and banks. Large woody debris can form log-step pools and lateral scour pools in these channels [33]. Across process groups a variety of stream channel sizes are represented ranging from channel bed widths of 1.9 to 50.1 meters.

**Table 1. Candidate predictors in linear mixed models, LMMs.** Predictor names are those used in the manuscript text; abbreviations in parentheses are used in tables and figures.

| Predictor (abbreviation) | Description | Data source |
|---|---|---|
| *Random Effects* | | |
| Watershed (WS) | HUC12 watershed name to account for spatial nesting of sites within watersheds | National Hydrography Dataset https://www.usgs.gov/core-science-systems/ngp/national-hydrography |
| Site | Site identifier to account for correlation attributable to repeat surveys at sites. | |
| Year (YR) | The year a survey was conducted to account for temporal autocorrelation | |
| *Intrinsic* | | |
| Process Group (PG) | Stream channel classification based on hydrologic function, landform, and channel morphology. Classes combined into moderate gradient, floodplain, alluvial, and high gradient. | Paustian et al. 1992 |
| Subwatershed Area (CatArea) | Natural log of subwatershed drainage area ($km^2$) using downstream site boundary as a pourpoint | Digital elevation model (DEM). 1-m lidar DEM for Prince of Wales Island, 5-m IfSAR DEM for rest of study area |
| *Management* | | |
| Old Growth Forest (OldGrowth) | Proportion of the subwatershed area in size class 4 forest (>22.9 cm [>9 in] diameter at breast height or 137 cm [4.5 ft] above the ground) and defined as "old-growth sawtimber" >150 years old | Covertype layer U.S. Forest Service (USFS) Region 10 |
| Canopy Cover (Canopy) | Proportion of the subwatershed area in any type of forest. | Covertype layer USFS Region 10 |
| Pourpoint Subwatershed Harvest (PPHarv) | Proportion of subwatershed area that had been harvested prior to the habitat survey date | Combination of harvest data from USFS and other landowners, such as State of Alaska, Sea Alaska Corporation, and Alaska Department of Natural Resources |
| Riparian Harvest (RipHarv) | Proportion of a reach's stream analysis buffer (SAB, 91 m wide on each side of the stream) harvested prior to the habitat survey date | Overlap of harvest data (see above) with 91-m stream analysis buffer applied on each side of surveyed stream reach. |
| Harvest Lag (HarvLag) | Number of years between the date of the largest buffer or pourpoint harvest event for a given site and the date of that survey | |
| Road Crossings (RDCrossDens) | Number of intersections of road and stream layer files in a subwatershed divided by subwatershed area | System roads from the Tongass National Forest Transportation Database and Alaska Hydro Stream Lines. |
| Road Lengths (RDLenDens) | Length of road in a subwatershed divided by subwatershed area | USFS Tongass National Forest Transportation database—System Roads |
| Restored (Rest) | Binary variable for whether restoration in the form of large wood addition occurred prior to a survey. Over 99% of restoration actions were restoring large wood to stream channels | |
| *Climate* | | |
| Hydrologic Regime (HydroClass) | Broad scale hydrologic classifications for each survey site based on dominant precipitation type: rain (includes transitional rain–snow regimes) vs. snow | Sergeant et al. 2020. |

## Forest management

Disturbances can drive eco-physical responses at multiple scales, making multi-scale analyses important in understanding ecological processes [38]. Therefore, we consider the effects of harvest at multiple spatial scales by calculating the amount of prior cumulative forest harvest and the largest harvest event for each survey at the subwatershed scale and within the SAB. Stream habitats respond in a variety of ways to timber harvest in the upstream subwatershed and in the riparian area surrounding each sample site, including persistent shifts in mean values or trends over time [39]. Therefore, we included a lag metric for time since harvest that could interact with the amount of harvest at the subwatershed and SAB extents. We also considered a quadratic relationship with harvest lag based on the hypothesis that remnant slash and riparian windthrow may increase wood densities during episodic events after harvest but later gradually decline owing to reduced supply from regenerating forests.

In addition to forest harvest, we used the most recent GIS layer for forest-stand condition, provided by the Tongass NF, to calculate the area of productive old-growth forest (i.e., the proportion of a watershed with trees estimated at >23 cm (9 in) dbh and over 150 years in age) and canopy cover (i.e., forest of any kind) at the subwatershed scale. We also summarized the length of roads and the number of road crossings at the subwatershed scale. Instream restoration (usually the addition of large or key wood pieces) has been completed at many sites across the Tongass NF. Exploratory analyses in objective 1 found that stream habitat at restored sites could differ from non-restored sites. Therefore, we evaluated a binary variable for whether a site had experienced in-stream restoration (i.e., wood restoration prior to the survey date for objective 2; we excluded restored sites from the trend analyses in objective 3).

## Climate data

To capture differences in stream habitat that could be related to hydrologic regimes, we used published hydrologic classifications of rain-dominated and snow-dominated developed for watersheds in the Gulf of Alaska based on a classification of stream flows from a complex run-off model that incorporates a digital elevation model, landcover dataset, glacier inventory, and soil characteristics [28]. Watersheds characterized as transitional between rain- and snow-dominated were grouped with the rain-dominated class owing to low sample size. The full dataset contained 514 sites with rain-dominated hydrology and 338 sites with snow-dominated hydrology. All predictor variable descriptions are provided in Table 1.

## Statistical analyses

In objective 1, we used multivariate analyses to test the hypothesis that stream-habitat conditions on the Tongass NF were significantly different among the broad management classes of "reference," "restored," and "harvested," (S1 Fig) as well as the floodplain and moderate-gradient process groups. We tested for differences in multivariate mean habitat conditions among management groups and channel-process groups using non-parametric, multi-response, permutation procedure (MRPP) simulation tests utilizing a Euclidian distance metric (R Vegan package, 999 permutations) [40]. Post-hoc pairwise comparisons between groups were conducted with the same method as multi-way comparisons, except that only two groups were tested at a time. We used the Bonferroni correction to adjust the significance level ($\alpha$) for multiple comparisons. MRPP provides a p-value and the chance-corrected, within-group, agreement statistic (A) as an effect size that ranges from −1 to 1, with values closer to 1 denoting greater intergroup difference. The effect size is important for interpreting the ecological significance of the test because sample sizes >100 can produce significant p-values with small effect sizes (i.e., A < 0.1) [41].

Habitat metrics included in our multivariate analysis were hydraulic radius, undercut bank density, width:depth, relative submergence, $D_{50}$, large wood density, key wood density, pool density, residual pool depth, residual pool depth/channel bedwidth, pool spacing, relative pool area, pool size, total pool area, and total riffle area. Multivariate analyses require complete data matrices, therefore we filtered out incomplete records leaving 346 surveys. We imputed missing values for undercut bank density for 7 survey records using the value from another survey of the same site that was closest in time. We overlaid group membership on a PCA ordination with metric loadings and site scores that graphically represented correlations of individual metrics with PCA axes and differences among sites in management classes and channel-process grouping.

For objective 2, we used univariate linear mixed effects models (LMMs) to explore how more complex relationships among intrinsic landscape characteristics, management actions,

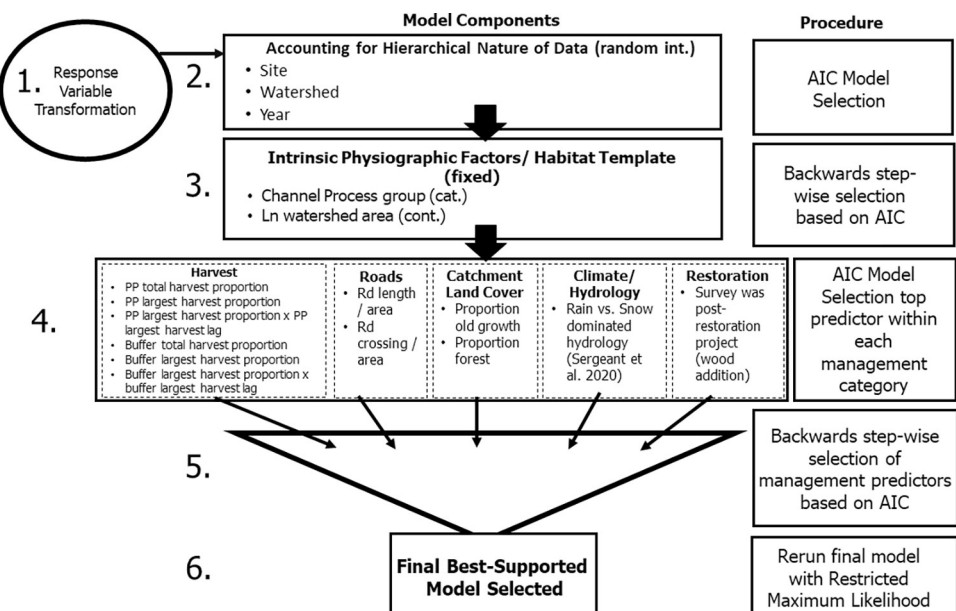

**Fig 2. The 6-step univariate linear mixed model process.** Abbreviations: rd (road), pp (pourpoint/subwatershed catchment), ln (natural log), cat. (categorical variable), cont. (continuous variable), AIC (Akaike Information Criterion). Refer to Table 1 and S1 Table in S1 File for variable abbreviations.

and climate-driven hydrology may affect stream-habitat metrics (see S3 Table in S1 File for model-specific sample size, *n*). We adapted stepwise modeling frameworks (e.g., Dunn et al. [42]) that have also been used to isolate the signatures of natural physiographic processes and the legacy of human disturbance in stream habitats in the Pacific Northwest [26, 27] into our own 6-step modeling process conducted in the Lme4 package in R (Fig 2) [43]. All continuous predictors were centered and scaled by mean and standard deviation.

In step one (see Fig 2), we examined the distribution of the response variable for departures from normality. Transformations such as natural log were implemented when necessary to better approximate normality (i.e., skewness <|1|). A small constant was added to all response variable values when zero values were present prior to log-transformation.

In step two (see Fig 2), we accounted for nuisance spatial and temporal correlation of surveys and repeated measures of sites with random intercepts for the following factors: site, HUC 12 watershed, and survey year. We selected the random-effects structure based on minimum Akaike Information Criterion (AIC) [44] among all possible combinations of these three variables. We screened models for lingering spatial and temporal autocorrelation effects by examining semivariograms.

In step three, we developed a base model to control for variation in the habitat metrics resulting from geomorphic processes and physiographic patterns. We started with a global model containing four intrinsic factors and three random effects listed previously (Table 1) that set a template to constrain ranges of variability in stream habitat [45]. Next, we eliminated non-informative predictors by backward-stepwise model selection using restricted maximum likelihood methods that are required when comparing likelihood among models with different fixed effects [46]. In this process, we began with a global model containing all intrinsic variables, which was compared to the suite of models in which a single variable was dropped. AIC was calculated for each model and then the least-supported variable was dropped and removed from further steps. The process repeated until no more variable removals were supported according to minimum AIC.

In step four, we added management variables in five categories: timber harvest, roads, land-cover, hydrological classification, and restoration history (Fig 2). Hydrologic classification was included with other management variables because of the role of anthropogenic activities such as greenhouse gas emissions in driving ongoing climate change across the Tongass NF. Variables were added individually, and the top supported variable based on minimum AIC was retained from each category for use in step five.

In step five, we combined the base intrinsic landscape model and added the top management/climate predictors from all categories selected in step four. We screened variable pairs for collinearity by identifying those with variance inflation factors >5 and pairs of variables with pairwise Pearson correlation coefficients >0.7 and removed the least supported of the pair. Next, we performed the same backwards selection procedure from step two on management and climate variables only.

In step six, the final model was refit using restricted maximum likelihood (REML) procedures that produce unbiased estimates of variance components [39, 46]. The parameter coefficients from the top models for each response metric were assessed individually to calculate effect sizes and determine relationships between management actions, climate, and stream habitat. We report conditional and marginal $R^2$ for the final model, which measures the amount of variation explained by random- and fixed effects, respectively [47], as well as partial $R^2$ for select predictors using the partR2 package in R [48].

We calculated marginal effect sizes with our top models in order to illustrate the magnitude of expected changes in habitat metrics in response to predictor variable changes. To do this we predicted the percent change in the habitat metric response variable to a 25% change of the observed range of a given continuous management or intrinsic predictor variable while holding all other continuous predictor variables constant at their mean. For binomial predictors such as restoration status or hydroclass, we predicted a percent change in response variables from one level to the other.

For objective 3, we examined potential trajectories in stream habitat conditions by isolating the effect of a temporal variable (i.e., survey year). Because of the low number of repeated surveys within stream reaches, we implemented a space-for-time approach, while also controlling for spatial variation explained by natural landscape factors. The lack of site-selection randomization could have biased temporal trends if we were unable to control for confounding trends in site-selection criteria over time. Therefore, we decided to only examine trends for undercut bank density, large wood density, and key wood density. These metrics were measured during many of the same surveys, and studies suggest observer measurement precision is sufficient for these variables to detect modest change over time and space [49]. We divided sites into reference and harvested management classifications introduced in objective 1 and examined whether trends vary by hydrologic regime. Thus, rain- and snow-dominated hydrology could be captured at the subwatershed scale, as these hydrologic regimes may be changing differently in response to interactions between management and climate change. Restored sites were excluded from these analyses owing to low sample size. All possible combinations between the additive and interactive effects of year and hydrology were fit and the top model based on minimum AIC was retained for inference. We are not aware of any systematic temporal biases in site-selection criteria, selected site condition, or habitat survey training over time, but these factors have the potential to affect our results if present. Furthermore, we attempted to control for nuisance variation by using random and spatial and fixed physiographic effects and by separately analyzing trends in reference sites and harvested sites. We graphically present the marginal effect of survey year that may be indicative of temporal trends in the data by predicting the change in the response variable while holding all other continuous predictors constant at their mean. We also report conditional and marginal $R^2$ for the final model, which measures

the amount of variation explained by random- and fixed effects, respectively [47] using the partR2 package in R [48].

## Results

### Assessment of stream habitat differences in harvested, not harvested, and restored subwatersheds under the control-impact management classification framework

The MRPP analyses found significant mean differences among management classes (harvested, not harvested, and harvested then restored) and process groups (floodplain and moderate gradient). However, in pairwise comparisons of management classes, between-group differences had very small effect sizes (mean A = 0.027) [41]. Furthermore, only the restored sites were significantly different from the other two management classes, with no significant difference observed between harvested and reference classes after implementing the Bonferroni correction to α for multiple tests (S5 Table in S1 File). However, the MRPP analysis suggested slightly stronger differences existed between floodplain and moderate-gradient sites owing to the landscape level factors captured in our modification of Paustian's hierarchical channel process group system (A = 0.04 p < 0.0001).

In the PCA, the first two principal components collectively explained 45.2% of total variance. Axis 1 was positively correlated with key wood and pool densities (i.e., shorter pools) and negatively correlated with residual pool depth and pool area (i.e., fewer but longer pools) (Fig 3A). Axis 2 was positively correlated with $D_{50}$ and riffle area and negatively correlated with relative submergence (Fig 3A). There was a substantial overlap of the distribution of survey scores for the first two principal components among the three management classes, corroborating the small effect sizes from the MRPP (Fig 3B). The minimum convex polygons (defined from the outer edges of the data points in each group) for harvested sites were slightly larger than the other groups (Fig 3B), potentially suggesting that the variability in disturbance histories for harvested sites resulted in greater habitat variability relative to reference sites. PCA Axis 1 explained much of the difference between the floodplain and moderate-gradient process groups. Floodplain process groups were associated with greater residual pool depth, whereas moderate-gradient channels were associated with higher densities of key wood and pools (Fig 3C). The outer boundary of the minimum convex polygon for harvested sites extended further to the right on Axis 1 (Fig 3B), a pattern driven primarily by some small floodplain and moderate-gradient harvested sites that had greater densities of key wood, greater pool density, and larger pool area (Fig 3D).

### Variation in individual habitat metrics attributed to spatial, temporal, and intrinsic landscape factors

Spatially clustered patterns of geology and topography were apparent across the Tongass NF, with more similar characteristics for sites within watersheds. The proportion of the watershed classified as forest cover was high and relatively invariable across the study area (Fig 1H), reflecting regrowth of harvested areas. Most subwatersheds had high proportional area of forest cover (M = 92%, SD = 12.1); however, more variation existed in the proportion of old-growth forest cover (M = 42%, SD = 21.6), reflecting that past timber harvest has reduced the amount of old-growth forest (Fig 1F–1I). Historical timber harvest at both the subwatershed and riparian-zone scales was concentrated in easily accessible coastal subwatersheds throughout Prince of Wales and Kupreanof islands, or near settlements of Baranof and Chichagof islands (Fig 1A, 1F and 1G). Timber harvest was low for most mainland subwatersheds. The

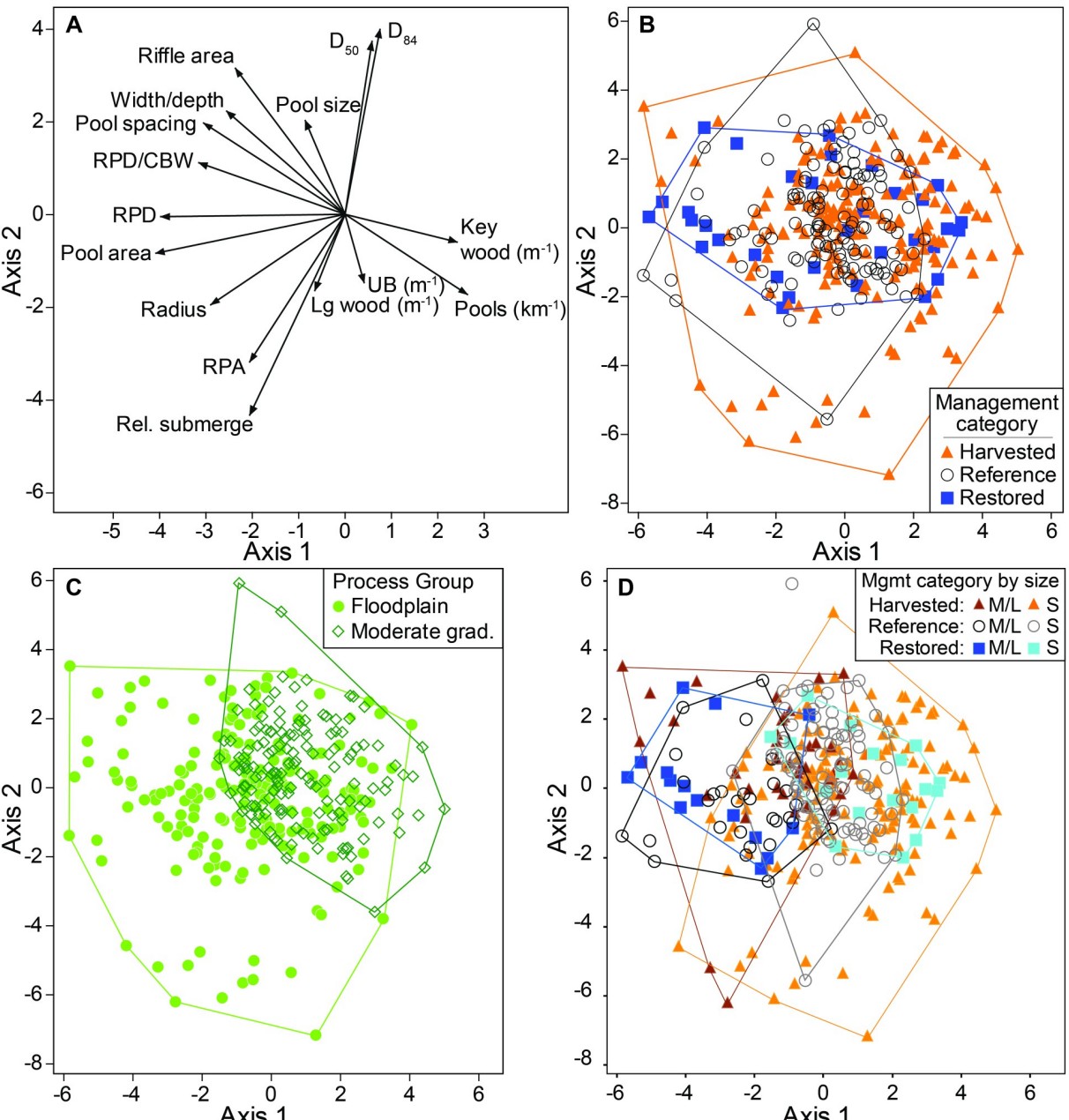

**Fig 3.** Principal components analysis biplots of 346 sites with full data matrices for all included metrics: A) vector loadings of individual metrics to show correlations along axes 1 and 2 (RPA- relative pool area, RPD- residual pool depth, UB- undercut banks, Rel submerge- relative submergence, Lg wood- large wood; CBW-channel bedwidth, additional variable information provided in S1 Table in S1 File), and individual sites plotted by their eigenvalues for axis 1 and axis 2 with sites color coded by B) management class, C) process group, and D) management class and relative channel size (small- "S" or medium/large-"M/L"), which is a part of the Paustian [35] classification system.

hydrologic influence of rain was greater in western lowland maritime-influenced subwatersheds and snow influence was greater in high-elevation island and eastern subwatersheds (Fig 1E). Road-length density had a similar spatial distribution as subwatershed timber harvest ($r = 0.71$) because most were timber harvest access roads. However, they were not simultaneously supported by model selection procedure and therefore never occurred in the same model.

Random effects controlled for extra variation attributable to repeat surveys of sites (10/10 models), surveys of sites nested within HUC 12 watersheds (10/10 models), and surveys nested within years (8/10 models) (S2 Fig). Variation explained by fixed effects of best-supported models was low (marginal $R^2$ = 0.03–0.28) relative to random effects (conditional $R^2$ = 0.51–0.93; S3 Table in S1 File). The greatest amount of variation in habitat metrics was explained by fixed intrinsic and management predictors for width:depth (marginal $R^2$ = 0.28) and residual pool depth (marginal $R^2$ = 0.19). In many models, subwatershed area explained the greatest proportion of variation among fixed predictors and on its own explained ~50% of the fixed effect variation in width:depth (marginal $pR^2$ = 0.14). In contrast, fixed predictors explained little variation in pool size (marginal $R^2$ = 0.03) or pool spacing (marginal $R^2$ = 0.09).

As expected, variation in stream habitat metrics was commonly associated with subwatershed area and process group (7 and 6 of 10 habitat metrics, respectively). Subwatershed area ranged from < 0.1 to 114.5 km$^2$ (Fig 1B) and had the largest effect size of any variable, with a 76% predicted increase in width:depth for a 25% increase in the observed range of subwatershed area (ln-transformed) (Table 2). Greater subwatershed area was also associated with greater pool spacing, residual pool depth, and relative submergence and lower pool- and key wood density. The overlapping 95% confidence intervals of beta coefficients for the floodplain

**Table 2. The coefficients (Beta), P values (P), and predicted marginal effect sizes (% change) of continuous or binary explanatory variables.** The unit column reports whether the marginal effect size presented was for a 25% change in the observed range of a continuous predictor variable or a change from one level to the other of a binary variable. The change intervals (Lower; "LPI") and (upper "UPI") are 95% bootstrapped prediction intervals accounting for random effects and thus are wider than confidence intervals for the mean response. Bold type indicates positive predicted percent change in the response variable for increase in a predictor variable, whereasor italic type indicates negative predicted changes.Refer to Table 1 and S1 Table in S1 File for variable abbreviations.

| Response | Predictor | Unit | Beta | P | Change | Change 95 LPI | Change 95 UPI |
|---|---|---|---|---|---|---|---|
| UB | PostRest | Bin | -0.31 | <0.001 | *−31.2%* | *−53.8%* | *−8.9%* |
| | BuffHarvMax | 0.25 | -0.12 | <0.001 | *−8.9%* | *−26.9%* | **7.8%** |
| Width:depth | CatArea | 0.25 | 0.28 | <0.001 | **75.8%** | **25%** | **150.7%** |
| | PPHarvestPropPre | 0.25 | 0.07 | 0.06 | **6.2%** | *−12.8%* | **28.3%** |
| | RdCrossDens | 0.25 | -0.06 | 0.040 | *−9.3%* | *−27.9%* | **19.1%** |
| Pools/km | CatArea | 0.25 | −9.53 | <0.001 | *−17,85* | *−35.78* | **0.35** |
| | PostRest | Bin | -12.83 | 0.009 | *−12.82* | *−25.58* | **-0.56** |
| PoolSpace | CatArea | 0.25 | 0.25 | <0.001 | **60.2%** | *−16.7%* | **197.3%** |
| | PostRest | 0.25 | 0.75 | <0.001 | **111.9%** | **38.5%** | **238.9%** |
| RPD | CatArea | 0.25 | 0.09 | <0.001 | **18.5%** | **4.0%** | **33.0%** |
| | PostRest | Bin | 0.05 | 0.037 | **5.2%** | *−4.6%* | **15.4%** |
| | RDLenDens | 0.25 | -0.02 | 0.037 | *−2.5%* | *−11.7%* | **8.2%** |
| PoolSize | RdCrossDens | 0.25 | -0.07 | 0.008 | *−10.8%* | *−31.7%* | **9.7%** |
| LW/m | HydroClass | Bin | 0.13 | 0.033 | **16.5%** | *−0.8%* | **30.2%** |
| | OldGrowth | 0.25 | 0.05 | 0.041 | **6.5%** | *−7.1%* | **20.9%** |
| KW/m | CatArea | 0.25 | -0.15 | <0.001 | *−24.5%* | *−50.2%* | **20.6%** |
| | HydroClass | 0.25 | 0.23 | 0.048 | **25.3%** | **1.6%** | **55.8%** |
| | OldGrowth | 0.25 | 0.08 | 0.065 | **10.1%** | *−10.7%* | **38.3%** |
| RelSub | CatArea | 0.25 | 0.10 | 0.007 | **22.5%** | *−24.8%* | **70.0%** |
| | PostRest | Bin | 0.20 | 0.067 | **20.4%** | *−6.5%* | **45.9%** |
| | PPHarvMax | 0.25 | 0.10 | 0.047 | **12.7%** | *−22.4%* | **43.9%** |
| D50 | CatArea | 0.25 | 2.75 | <0.001 | **5.68** | *−3.31* | **14.22** |
| | PostRest | Bin | -3.62 | 0.14 | *−3.62* | *−8.68* | *−1.73* |
| | RDCrossDens | 0.25 | -1.28 | 0.15 | *−5.48* | *−13.13* | **2.40** |

and moderate-gradient group terms, which comprised a majority of the dataset, suggests that they were not statistically different (S3 Fig). Nonetheless, AIC-based model selection supported the inclusion of process group in several models, evidence that this variable may control for variation in stream habitat metrics, especially for process groups with lower sample sizes such as in the alluvial and high-gradient process groups.

## Variation in habitat metrics attributed to management legacies

Best-supported models for pool-metrics and $D_{50}$ included few management variable effects (Table 2 and S3 Table in S1 File). In contrast, best-supported models for large or key wood densities contained forest management predictors such as prior riparian harvest and proportion of old-growth cover within the subwatershed. However, these fixed effects had relatively small effect sizes (Table 2). Fixed effects from multiple predictor groups explained channel morphology metrics and relative submergence.

The most consistently influential management predictor was restoration, which was included in best-supported models for 6 of 10 habitat metrics (S1 Fig). Restored sites had higher pool spacing, lower densities of undercut banks and pools, lower residual pool depth, and lower $D_{50}$. In contrast, there were no differences in large and key wood densities based on restoration status.

Effects of harvest variables were noted at the subwatershed scale for two metrics and at the SAB scale for two other metrics. Harvest at the subwatershed scale was positively correlated with width:depth and relative submergence (i.e., channel widening, Table 2). SAB harvest was negatively correlated with undercut bank density. There was a complex relationship between large wood density and the interaction between proportional riparian harvest and a quadratic term for time since harvest. The model predicted higher large wood densities at sites with large riparian harvests that occurred >40 years prior to the survey compared with sites without riparian harvest or with recent riparian harvest (Fig 4).

Proportion of subwatersheds in old-growth forest cover is another sign of lack of recent harvest, and was primarily associated with wood-related metrics. Both large wood and key wood densities increased with increasing proportions of old-growth forest. No relationships were identified between subwatershed canopy cover and habitat metrics.

Road-crossing density was negatively associated with pool size. Road-crossing density and road density were retained in models for width:depth and residual pool depth, respectively. However, the effect sizes were small, with prediction intervals that were nearly symmetric around 0 (Table 2).

Sites with a rain- or snow-dominated subwatershed hydrology did not differ for metrics describing channel morphology and sediment transport that may be predicted to respond to hydrologic variability. Instead, sites in snow-dominated subwatersheds had 16.5% greater large wood densities and 25% greater key wood densities than those in rain-dominated subwatersheds (Table 2).

## Space for time analysis of stream habitat trends across the Tongass NF

Sample sizes of surveyed sites varied across habitat metrics in the reference, harvested, and combined datasets, which influenced statistical power available to detect temporal trends. Key wood ($N_{combined}$ = 784, $N_{reference}$ = 370, $N_{harvested}$ = 343) and large wood ($N_{reference}$ = 357, $N_{harvested}$ = 338) had the largest sample sizes, and 98% of the measurements of large wood density had a paired measurement of key wood density in the same survey. In contrast, most other metrics had data spanning the period from approximately 1996 to 2020 (Fig 5); however, undercut banks were measured at few harvested site surveys prior to 2010.

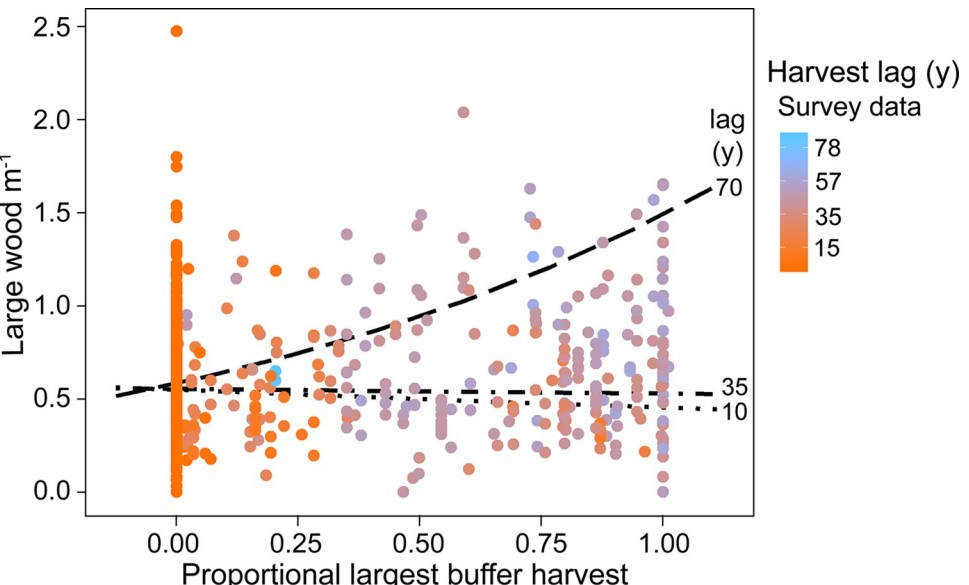

**Fig 4. Predictions of the marginal effects of the interaction between the largest proportional riparian harvest and a second-order term for years since harvest on the response variable large wood density (large wood per length of surveyed reach in meters).** Relationships are illustrated for three different time spans of harvest lag, namely, 10, 35, and 70 years post-harvest. The raw survey data are displayed by the points with color representing the time between harvest and survey years. The figure illustrates no predicted change in large wood densities in surveys ≤35 years post-riparian harvest for any magnitude of riparian harvest. However, sites surveyed >60 years following a large riparian harvest had higher predicted densities of large wood.

Year was retained in all nine of the best-supported models (3 metrics x 3 datasets), showing support for temporal trends (Table 3). Interactions between the hydrologic regime and year variables allow trend slopes to differ for rain- versus snow-dominated sites and were supported in three of nine models (Table 3). Temporal patterns were generally similar among analyses conducted with the combined, reference, and harvested datasets. Notably, there were decreasing densities of key wood in all datasets except for snow-dominated sites in the harvested dataset.

The direction of metric trends differed between rain- and snow-dominated hydrologic regimes in only 1 of 9 best-supported models. Predicted effect sizes from best-supported models include increases in undercut bank densities of 26 to 97% over 10 years for the reference, harvested, and combined datasets. Contrasting trends emerged for large wood and key wood densities in the reference and combined datasets. Large wood densities increased by 13 to 34% and key wood densities decreased by 14 to 25% (except at snow-dominated sites in the harvested dataset, which increased 43%; Table 3).

Random intercepts revealed that spatial correlation for repeated measures at the site scale and for sites within the same watershed were important components in the variation structure. For example, a large proportion of variation in large wood density was explained by the random intercept for site in the combined and reference dataset (37–38%; Fig 6). However, more variation was explained by the watershed random intercept in key wood density trends in the combined (20%) and reference (27%) datasets relative to the harvested dataset (9%). The amount of variation explained by intrinsic landscape, hydrologic regime, and trend fixed effects was relatively low for all trend analyses (marginal $R^2$ = 0.06–0.32). The main effect of the year fixed effect explained the greatest proportion of variation for undercut banks (marginal $pR^2$ = 0.08–0.12).

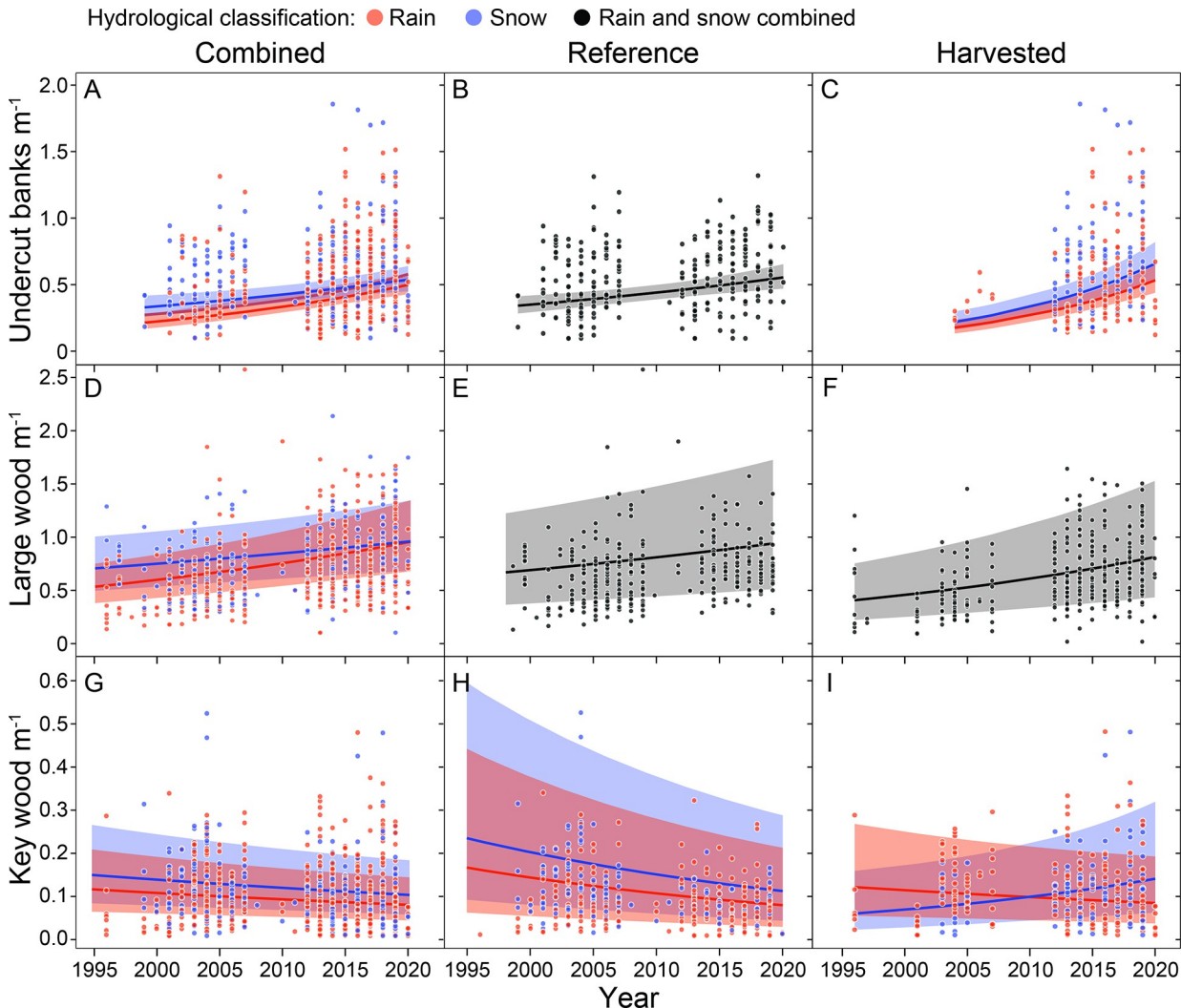

**Fig 5. Predicted time trends for three response variables, with raw data points, and trend lines are for rain (red), snow (blue), or combined (black) hydrologic classifications.** Plots are in grayscale where a difference between rain and snow hydrologies was not supported for that dataset and response variable combination. Plot rows are organized by response variable, with panels A., B., and C displaying the predicted marginal effect of year on undercut banks m$^{-1}$ by year, plots D, E, and F the marginal effect of year on large wood m$^{-1}$, and plots G, H, and I the marginal effect of year key wood m$^{-1}$. The columns represent the dataset used in the model. The first column (combined) was fit using all available habitat surveys (including restoration sites with wood additions), the second uses only reference sites, which are defined as having <6% pourpoint subwatershed harvest, and the third column (harvested) uses all sites with >6% pourpoint subwatershed harvest regardless of whether the harvest was riparian or buffered but excluding restored sites that had pre-survey wood additions.

## Discussion

Our exploration of a large, 30-year, forest-wide stream habitat dataset using type-3 analysis [24] revealed patterns in stream habitat metrics associated with spatial and temporal complexity in intrinsic landscape factors, forest management practices, and climate-driven hydrologic regimes. First, we found weak support for identifiable differences between previously investigated broad reference, harvested, and restored management classes that have been used to guide stream habitat management in the Tongass NF. The MRPP analysis revealed slightly stronger support for differences in channel process groups, such as floodplain compared with moderate-gradient channels. Therefore, we reanalyzed the data to explore nuanced relationships between intrinsic landscape factors and human actions associated with forest

**Table 3. Marginal effect sizes and their 95% bootstrapped prediction intervals of a 10-year change in survey year for five habitat metrics.** Hydroclass denotes rain- or snow-dominated hydrologic regimes. Where predicted effects are the same for both rain and snow, the model selection supported models with no interaction between hydroclass and year predictors. The landscape predictors column shows the natural landscape-level variables that were used to control for nuisance variation in response attributable to intrinsic watershed characteristics. See Tables 1 and 2 for variable abbreviations. Positive relationships are in bold face, negative relationships are in italics.

| Response | Landscape predictors | Hydro class | Combined | Reference | Harvested |
|---|---|---|---|---|---|
| UB | None | Rain | **49.4 (29.2–72.4)%** | **25.6 (11.7–41.1)%** | **97.1 (56.0–147.7)%** |
| | | Snow | **26.2 (10.3–45.3)%** | **25.6 (11.7–41.1)%** | **97.1 (56.0–147.7)%** |
| LW m$^{-1}$ | PG+ | Rain | **26.2 (15.6–37.1)%** | **16.3 (2.5–31.2)%** | **33.9 (15.7–55.0)%** |
| | | Snow | **13.0 (1.2–24.7)%** | **16.3 (2.5–31.2)%** | **33.9 (15.7–55.0)%** |
| KW m$^{-1}$ | CatAreaKM+PG+ | Rain | *−13.5 (−26.3–2.8)%* | *−25.3 (−3.9− −42.1)%* | *−14.0 (−30.2−6.2)%* |
| | | Snow | *−13.5 (−26.3–2.8)%* | *−25.3 (−3.9− −42.1)%* | **42.7 (5.7–90.3)%** |

management and individual stream habitat metrics that may be targeted for improvement during restoration. Our results reaffirmed that catchment area (through discharge magnitude), and to a lesser effect, process group, are dominant factors in establishing a template for the formation of stream habitat features. Forest management-related predictors explained less variation, but several relationships had strong statistical support. Some major takeaways were intuitive, and others were surprising, including: (1) Sites with restoration post-harvest were significantly different from non-restored sites across a variety of metrics; (2) pool-related features respond slowly to forest harvest and post-harvest restoration compared to more responsive habitat features such as wood densities; (3) Forest stand age is more important than total forest cover at watershed scales in explaining wood densities in streams; (4) spatial and temporal dimensions at which harvest has occurred (riparian vs. watershed, lag since harvest) are important considerations that were overlooked in previous habitat assessment designs. Finally,

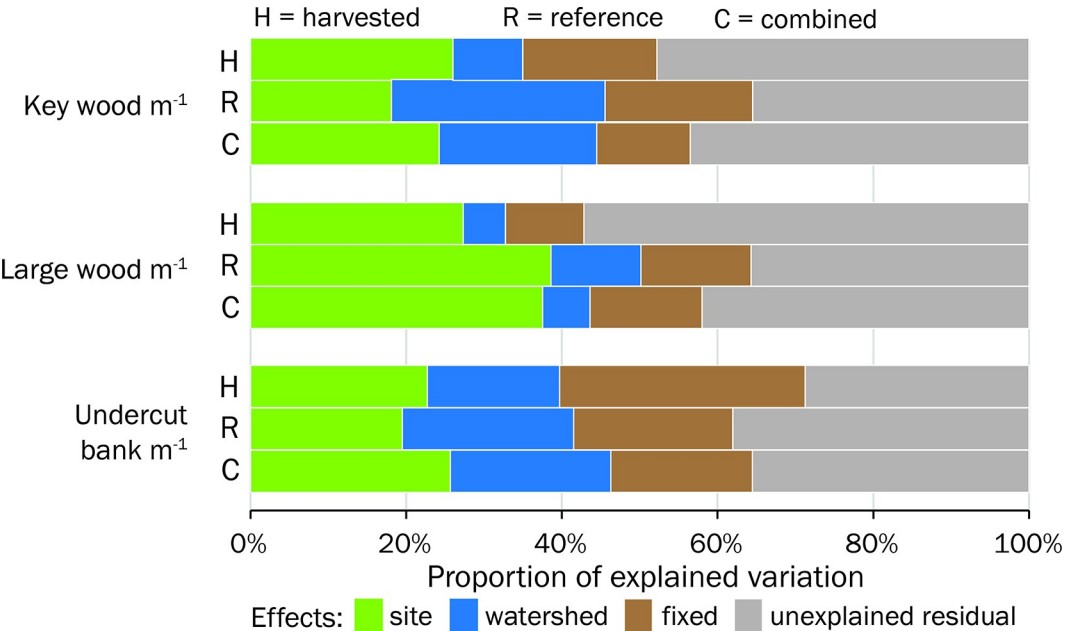

**Fig 6. Proportion of variation explained by site and watershed random effects and fixed effects, in addition to the residual unexplained variation for best-supported trend models.** The fixed effects portion includes any intrinsic landscape predictors, and if included, year and HydroClass. C: models for three response variables using the unfiltered dataset. R: models constructed using only data from reference sites with <6% subwatershed harvest and no Stream Analysis Buffer (SAB) harvest. H: models constructed using only data from harvested sites with >6% subwatershed harvest or >0% SAB harvest.

temporal trends may foreshadow future trajectories including the concerning decline in vital channel-shaping key wood densities. Future experimental studies could inform our understanding of the mechanistic processes underlying these correlative patterns.

## Assessment of broad management classifications as a means to characterize variability in stream habitat condition

The MRPP analysis and PCA ordinations showed greater separation in stream habitat characteristics based on natural landscape-scale variation described by process group than by broad categories of reference, harvested, or restored management classes. The dataset depicted a spatially heterogeneous forested landscape produced by a mosaic of past forest disturbances, where traditionally applied binary classifications of human management intensity (i.e., "reference" vs. "harvested" [7]) may be becoming a less useful paradigm for guiding management decision-making. A more complete picture of landscape variability required the nuanced consideration of management types, intrinsic landscape-scale characteristics, and time. The analysis showed that smaller, higher-gradient channels are predisposed to retaining key wood and forming a greater number of small pools, whereas larger floodplain streams have fewer but larger pools. This complexity in landscape, management, restoration, and climate effects transcends what can be described by categorical process group and management history variables alone and was a primary motivation for us to explore continuous variables describing new aspects of the Tongass NF landscape template (i.e., subwatershed area), specific types of management intensity (i.e., riparian or watershed harvest, and road construction), time for post-harvest forest regeneration, and spatial position of harvest relative to stream channels.

## Role of the landscape template

In objective 2, our linear mixed model results inform our understanding of the landscape characteristics that influence stream habitats. Landscape characteristics also affect site accessibility and forest productivity for timber harvest and may be reflected in the spatial structure of observed forest management predictors. Our models showed effects of subwatershed area and process group for most metrics, which integrate factors correlated with stream power (i.e., stream discharge, valley form, and slope) that have explained the greatest proportion of variation in stream habitat in previous studies in Washington and Oregon [26]. We found that $D_{50}$ is not strongly associated with management activities that deliver fine sediments to stream channels with only a weak association with road crossing density (S3 Table in S1 File). Suggesting that sufficient stream power may exist to flush the bed surface of fine sediments [50]. Interestingly, the best-supported model for pool spacing still included subwatershed area despite the equation for pool spacing already adjusting for channel bed width (S3 Table in S1 File) [16]. This suggests that pool-spacing metric may have unexplained variation related to watershed size.

Pool density and substrate transport may be heavily constrained by intrinsic landform characteristics on the Tongass NF. Similarly, other studies observed only weak relationships between management variables and pool densities [26, 27]. In contrast, Beechie and Sibley [51] found that greater wood densities could force a reduction in pool spacing and an increase in pool area that was dependent on channel slope. In our dataset, 55% of surveyed reaches may have an abundance of wood-forced pools given that they had pool spacing values <3 [16].

## Management acting upon the template

Effect sizes of forest management variables on stream habitat metrics appeared generally small, but this may be attributable to changes in forest management practices over time and that

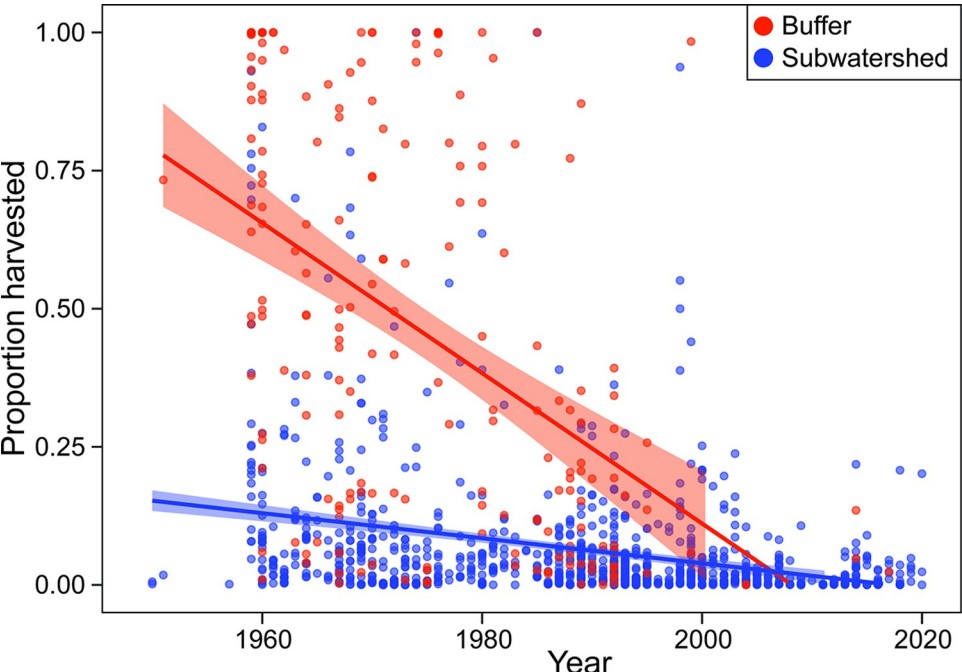

**Fig 7. Harvest trends associated with surveyed streams in our dataset across the Tongass National Forest.**
Subwatershed points indicate the proportion of a study site's subwatershed that was harvested during an individual harvest event. "Buffer" points are the proportion of a site's SAB (91-m-wide buffer on each side of the stream) that was harvested in an individual riparian harvest event. Harvest events are defined as the amount of harvest in a single year. Best-fit linear regression lines are displayed to illustrate declining harvest magnitudes over time at both spatial scales.

numerous regulatory mechanisms to protect stream habitats are working as intended on the Tongass NF. Strong declines in the harvest area at the subwatershed and RMA scales over the past 80 years, driven by market and regulatory factors (Fig 7), with very few harvest events having occurred inside the RMA or exceeding 25% of a surveyed site's subwatershed area since 1990.

The largest effect sizes occurred for restoration status, highlighting it as a primary factor to consider when assessing stream habitat conditions. Some observed effects may result from non-random selection procedures for restoration sites, rather than restoration outcomes *per se*. Stream restoration projects were nearly always completed in large lowland reaches with historical riparian harvest with documented habitat deficiencies. This may suggest that managers were able to effectively identify and prioritize restoration actions. Both large and key wood densities were similar in restored and unrestored sites because of wood additions during restoration that overcame key wood deficits. Without active restoration, it may take hundreds of years for riparian forests to regenerate and supply wood large enough to be recruited and retained in these wide, powerful lowland channels in SE Alaska [52, 53]. However, restored sites still had lower pool and undercut bank densities, and greater pool spacing, features that are important for overwintering juvenile salmonid survival [16]. This suggests that some habitat attributes like wood counts may be recovered instantly (coincident with the placement of wood during restoration activities), whereas others may change gradually over time [39, 54]. In our study, the average lag time from restoration to habitat survey was only 2.6 years. Sediment retention and scour processes may not have had adequate time to develop pools after wood installation, and documenting changes may require ongoing monitoring. Furthermore, some wood additions aim to stabilize existing pools rather than create new ones, and thus may not change the number of pools at a given site.

Many studies have found decreased wood densities in stream reaches located in harvested watersheds compared to those with no harvest [52]. However, time since harvest is also important, with Murphy and Koski [55] reporting large wood densities in streams declining to a minimum 70 years following riparian timber harvest. In contrast, our models found no effect of SAB harvest on large wood densities at short lag times but found higher large wood densities in reaches with large SAB harvest events 40–80 years before the survey date. It should be noted that because most riparian harvest occurred before 1990 and most habitat surveys occurred after 1990, there were only 78 surveys that had riparian harvest lags between 1 and 30 years. As a result, the dataset provides limited information to assess wood density response <40 years after a riparian harvest event. In headwater streams of SE Alaska, debris dams composed of small pieces of wood can form following riparian clear cuts that may initially increase wood densities [50]. However, this does not explain the lagged increase we observed, because these debris dams typically decompose rapidly and are flushed within 40 years [56]. More plausibly, mortality of red alder (*Alnus rubra*), a species common in disturbed riparian environments on the Tongass NF, grows quickly, reaching the threshold diameter for being considered LWD (>0.1 m) in <25 years. Alder mortality peaks at 90 years old, with the largest accumulations of hardwood woody debris occurring 60–80 years post-harvest, thereby aligning with the temporal relationships we observed [57].

Similarly, key wood densities were positively associated with the proportion of old-growth forest, suggesting that periodic recruitment of old-growth trees to stream channels maintains key wood densities. However, no relationships emerged for key wood with time since harvest. Possibly riparian regeneration 40–80 years following harvest had not yet produced trees large enough to meet key wood thresholds (these vary with stream size). These two findings align with other studies that emphasize the importance of conserving old-growth forests as sources of key wood in streams because the regeneration of old-growth forest stands can take centuries. On the other hand, we did not find a decrease in key wood following riparian harvest. However, legacy coniferous key wood pieces may have a long residence time that predate harvest [57]. Furthermore, the legally mandated 33-m RMB for fish-bearing streams on the Tongass NF did not provide total protection of the 91-m-wide SAB we used in analyses. Partial harvests of the SAB (post-1990) may have left narrow old-growth buffers vulnerable to windthrow providing temporary supplemental sources of key wood recruitment for streams [39]. Nevertheless, riparian harvests could reduce key wood recruitment in the long term as mature riparian timber supplies dwindle so that streams have incurred a key wood debt in their riparian forests from past harvests that have yet to be paid. Finally, the negative association between riparian harvest and undercut bank density aligns with previous studies that have also found fewer undercut banks in harvested habitats and suggest that riparian root mats can provide structural reinforcement [35, 58].

Our results align with previous studies that showed stream channel widening relative to depth that may be caused by hydrological changes such as increased discharge peaks following harvest [10]. After 1990, most harvest events on the Tongass NF were <30% of the subwatershed area. Harvest events before 1990 encompassed on average 9.5% of the subwatershed area, declining to 3.8% post 1990, and few harvests within the SAB recorded post-1990 (Fig 7). Smaller harvest areas and increased protection of riparian buffers may explain the low effect sizes of harvest variables on channel-morphology metrics. For example, in a previous study, a 30% subwatershed harvest resulted in a modest annual increase in discharge of 14–20%, with changes persisting for 10–20 years [59]. These results may support the effectiveness of more restrictive harvest regulations and practices, such as the Tongass Timber Reform act of 1990 in conserving stream habitat on the Tongass NF.

The importance of old-growth forest cover but not canopy cover at the subwatershed-scale in explaining large and key wood densities highlights that stand age and species composition may affect stream habitat. Even though some studies suggest 90% of the wood inputs to streams originate within <20 m of the stream channel [60], not all contributions are necessarily from local riparian stands. Episodic landslides can contribute to instream wood from upslope stands [61], and wood pieces may be transported as far as 2.5 km downstream in rivers of the Tongass NF [62]. Therefore, old-growth forests beyond the local riparian scale benefit stream habitat complexity, which is important to fishes.

To our knowledge, this is the first study to examine wood densities based on snow- vs. rain-dominated hydrologies and therefore there is limited previous research to help explain why snow-dominated hydrologies may have more large wood and key wood compared with rain-dominated systems. Surprisingly, climate-related hydrologic regime was not associated with channel morphology or sediment transport-related metrics. It is possible that snow-dominated subwatersheds with greater capacity to store frozen precipitation may experience less extreme autumn and winter flooding and, therefore, less flushing of wood, allowing for wood to accumulate in these systems.

Our results demonstrated decreased pool quality, as measured by pool size and residual pool depth, may relate to road density within a subwatershed, aligning with studies that found lower residual pool depths in streams with road construction, timber harvest, livestock grazing, mining, and other forms of management [56]. Timber harvest access roads can facilitate overland runoff resulting in fine-sediment transport that can fill in pools [12]. Although road crossings can be sources of sediment leading to aggradation in low-power channels they can also be barriers to sediment transport if culverts are present, leading to downstream channel entrenchment or decreased width to depth ratios, potentially explaining the negative association between width:depth and road-crossing density.

## Habitat trends informing possible future condition

Pervasive effects on aquatic habitats resulting from underlying hydrologic processes that are currently changing may be important to consider when describing variation in habitat conditions on the Tongass NF [21]. Previous studies of environmental change in Alaska freshwater environments have mainly focused on dimensions of water quality and episodic disturbances, with a more limited focus devoted to trends in physical habitat that could change in parallel with hydrology. The question of whether floods or drought fueled by climate change may act as temporary pulse disturbances for aquatic biota or whether they may inflict lasting press responses to stream ecosystems remains unanswered [63].

Temporal trends for undercut bank, large wood, and key wood densities may portend changes in habitat complexity that multiple life stages of resident and anadromous species depend on [64, 65]. Perhaps the strongest evidence for temporal trends in habitat metrics was the contrasting trajectories of large and key wood, because both models are built with mostly the same survey data. Although the increase in large wood is somewhat encouraging, the decline in key wood over time is concerning. Complex low-velocity habitats are crucial for the life stages of some salmonids and decreasing key wood. Without active restoration through the installation of key wood in the channel, juvenile salmonid habitat may decline into the future.

In contrast, undercut bank density is an additional component of habitat complexity and the increasing trend we observed may partially offset losses in low-velocity refugia and cover from predation provided by key wood that supports salmonid populations [66]. The mechanism for increased undercut bank density requires additional investigation. Increased undercut banks in Coastal Range streams in Oregon have been linked to increased discharge in

streams following watershed harvest with protected riparian buffers [67]. As previously discussed, larger runoff events on the Tongass NF may be related to climate change as increased winter rain-on-snow events may lead to bank scour, with riparian protections allowing well-developed root mats to stabilize undercutting banks.

The trends for habitat metrics we observed at reference sites, similar to those observed at harvested sites, cannot be attributed to changes in harvest practices because all reference sites had <6% historical timber harvest and no SAB harvest regardless of the date they were sampled. Comparison of habitat trends in rain- versus snow-dominated hydrologic regimes showed statistically detectable differences in trends for multiple metrics in combined reference and harvested datasets. The longitudinal sample size for this variable was more limited, and explanations for such a pattern are unclear but could signal shifts in scouring intensity in lowland rain-dominated systems. Future monitoring could track the effects of such environmental change.

## Considerations for the future management of northern coastal rainforests

Collectively, our results can help in planning future research, habitat monitoring, and management for aquatic habitats in northern coastal forests. The ordination analysis highlighted how increased variation in harvest practices and timing has increased the heterogeneity of conditions within harvested stream sites, underscoring the value of our examination of continuous harvest variables and lag interactions in this analysis. However, increased detail in data collection could improve our understanding of these patterns and processes underlying stream habitat condition. For example, rapid standardized assessments of riparian-zone condition or the use of emerging technologies with high-resolution imagery or lidar could document tree species composition, stand age, and evidence of vulnerability to harvest-induced blowdown that could help managers assess future wood recruitment and retention potential [68]. Some rearing habitat occurs in floodplain ponds but is not assessed in stream habitat surveys and may lie outside of buffered riparian areas. Inventory of other floodplain habitat features, such as beaver ponds, which have been shown to help sustain coho salmon recruitment [69], could also be incorporated into survey protocols. Large or key wood counts may be failing to capture information relevant to fish habitat. For example, Gomi et al. [56] found no significant difference in wood counts, but higher wood volume, in reaches flowing through old-growth forests compared to young forests. Measuring wood volume could be time consuming and prone to surveyor error, but surveyors may be able to classify large wood by species, wood type (i.e., conifer or deciduous), decay status, and stability to provide information on when wood recruitment occurred and to begin to flag streams at risk of future wood deficits.

Stream conditions may be changing independently from forest management in response to climate. Further, predicted changes in precipitation type could be severe for the region owing to the narrow thermal variability surrounding the freezing mark [70]. Flitcroft et al. [6] suggested that temporary decreases in young-of-the-year salmon populations in Tongass NF streams might have been related to extreme winter flooding that occurred in 2015. Synthetic stream models predict substrate scour [21] in response to shifts in hydrologic regimes and storm events. Such future conditions may make key wood an even more important habitat feature to maintain salmonid spawning gravels in high-power streams [21, 71]. Monitoring thereby may contribute to resilience of native salmonids.

Our work also has implications for habitat monitoring program design. The composite dataset that we used integrated data from multiple large-scale monitoring studies and is better suited for monitoring broadscale spatial patterns (e.g., [27]) rather than the mechanisms of habitat formation following disturbances (e.g., [56]), or temporal trends within specific

repeatedly surveyed sites (e.g., [39]). Ongoing broadscale monitoring, however, does not necessarily require additional sites. Rather, simulations show that sampling larger numbers of sites in a landscape-scale study usually does not significantly improve the power to detect trends [72]. Spatially balanced and stratified rotating-panel survey designs could be used to better monitor trends across reasonable spatial extents [73]. Further, the variance partitioning we conducted as part of our analyses (Fig 6) could be used in simulated power analyses (e.g., [72]) to design appropriate spatial extent (number of watersheds monitored), spatial resolution (number of sites within watersheds), stratification among different physiographic groups (geologies, topography, etc.) and temporal resolution (return rates) necessary to capture total variation across portions of the Tongass NF, although logistical constraints for accessing remote sites may still hinder its implementation. For example, the greater site correlation for pool spacing suggests that there is less variance in this metric within sites over time, and longer return intervals may be appropriate to detect change in this variable.

Finally, our results highlighting the legacy of harvest on key wood that structurally supports pool formation and provides cover for juvenile salmonids may inform future harvest regulations. Forest management has trended toward protection using stream buffers with smaller individual harvest events (Fig 7). The 2012 Tongass National Forest Land and Resource Management Plan outlined plans to shift away from old-growth harvest, with a goal of at least 50% of total harvest being second-growth harvest by the year 2027 [33], and recently President Biden issued an executive order directing agencies to catalogue and end the harvest of old-growth forests on federal lands [74]. Widespread second-growth harvest is unprecedented on the Tongass NF, and consequences for floodplain and moderate-gradient streams are uncertain. In faster-growing low-latitude forests with 40–80 year harvest rotations, reduced wood recruitment to streams has been documented, especially in large channels [52]. Short harvest return intervals may cut off long-term supplies of key wood even if narrow riparian buffers that are vulnerable to windthrow mortality are spared from harvest. The effects of second-generation harvest in high productivity, rain-dominated, marine-connected watersheds may disproportionately affect economically valuable species, including young coho salmon, which are more commonly associated with these streams, in contrast to resident cutthroat trout and sculpins (*Cottus* spp.), which are associated with interior snow-dominated systems [6].

## Conclusion

Our analysis of a long-term dataset of stream geomorphology and habitat in a federally managed temperate rainforest provides insights into broad-scale relationships between habitat condition, and past and present forest management practices, and climate-driven hydrology. Intrinsic and spatial variables explained most of the variation. The most important predictors of habitat metrics included stand age, harvest history, stream restoration among others. Surprising trends existed including increases in undercut banks and decreases in key wood densities that could be investigated further. Overall, our results show the influence of watershed-scale old-growth forests, suggesting that management for aquatic habitat may benefit from the assessment of forest stand condition at both riparian and subwatershed scales.

## Supporting information

**S1 File. Contains S1-S5 Tables with individual table captions as follows.** S1 Table. Stream habitat metrics that were used as response variables in multivariate and univariate statistical models and tests. Metrics are grouped into different categories (bold italic headings). Metric names are those used in the manuscript text; abbreviations in parentheses are used in tables and figures. S2 Table. Minimum size thresholds for wood pieces to be considered "key wood".

CBW- Channel Bed Width.S3 Table. Best-supported models for each variable. Refer to Tables 1 and 2 for variable abbreviations. "TF" specifies whether response variable was natural log transformed (LN) or not ("No"). N is the sample size of available surveys used in that model. $MR^2$ is the marginal pseudo $R^2$ representing the amount of variance explained by fixed variables and $CR^2$ and the conditional pseudo $R^2$ representing the amount of variance explained by the fixed and conditional variables. See S1 Table and Table 1 for variable abbreviation explanations. S4 Table. Descriptions of channel process group classes. S5 Table. Multi Response Permutation Procedure (MRPP) pairwise comparisons for three management classes and two process groups. A is the effect size and P is the p- value for the test. The significance threshold was adjusted for multiple comparisons using Bonferroni correction 0.05/4 tests = 0.013.
(DOCX)

**S1 Fig. Examples of stream sites in the "reference", "harvested", and "restored" management categories.** Panel A shows a stream in reference condition with no riparian buffer and limited subwatershed harvest. Panel B shows a site with substantial buffer and subwatershed harvest. Panel C shows a restored site that had buffer harvest and wood was installed to help improve stream habitat conditions. Collectively, these panels illustrate the multi-storied forest condition of reference stands, the conversion to Alder dominated riparian forest post-harvest that is discussed in the paper, and it shows some installed wood pieces at restoration sites.
(TIF)

**S2 Fig. Bar chart showing the number of best-supported univariate habitat models each predictor was included in out of 11 habitat variables that were modeled.** Bar position and x-axis labels divide predictors into 4 groups. Spatial/temporal random effects, intrinsic landscape fixed effects, management fixed effects, and climate fixed effects.
(TIF)

**S3 Fig. Beta coefficient estimates and their 95% confidence intervals for the two predominant process groups in the dataset (floodplain and moderate gradient) indicated by symbol type for seven habitat metrics where process group was retained in the best model.**
(TIF)

## Acknowledgments

Stream survey data were collected by hundreds of biologists, hydrologists and technicians employed by the Tongass National Forest and the Pacific Northwest Research Station. Thanks to every one of you for your diligence in support of aquatic resources. Special thanks go out to our mentors–Steve Paustian, Dan Kelliher, Julianne Thompson, Rick Woodsmith, Buck Bryant and Brenda Wright who laid the foundations that we walk upon. Thank you to Kathryn Ronnenberg for assistance in copy-editing and improving and formatting figures. Any use of trade, firm, or product names is for descriptive purposes only and does not imply endorsement by the U.S. Government.

## Author Contributions

**Conceptualization:** Michael J. Moore J., Rebecca L. Flitcroft, Emil Tucker, Katherine M. Prussian.

**Data curation:** Michael J. Moore J., Emil Tucker.

**Formal analysis:** Michael J. Moore J., Rebecca L. Flitcroft, Shannon M. Claeson.

**Funding acquisition:** Rebecca L. Flitcroft, Katherine M. Prussian.

**Investigation:** Michael J. Moore J., Rebecca L. Flitcroft, Emil Tucker, Shannon M. Claeson.

**Methodology:** Michael J. Moore J., Rebecca L. Flitcroft, Emil Tucker, Katherine M. Prussian, Shannon M. Claeson.

**Project administration:** Michael J. Moore J., Rebecca L. Flitcroft, Katherine M. Prussian.

**Resources:** Rebecca L. Flitcroft.

**Supervision:** Rebecca L. Flitcroft.

**Validation:** Michael J. Moore J.

**Visualization:** Michael J. Moore J.

**Writing – original draft:** Michael J. Moore J., Rebecca L. Flitcroft.

**Writing – review & editing:** Michael J. Moore J., Rebecca L. Flitcroft, Emil Tucker, Katherine M. Prussian.

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
