## [Decision Letter · Decision Letter 0]

27 Dec 2023

PONE-D-23-40089Same streams in a diff­­erent forest?  Investigations of forest harvest legacies and future trajectories across 30 years of stream habitat monitoring on the Tongass National Forest, AlaskaPLOS ONE

Dear Dr. Moore,

Thank you for submitting your manuscript to PLOS ONE. After careful consideration, we feel that it has merit but does not fully meet PLOS ONE’s publication criteria as it currently stands. Therefore, we invite you to submit a revised version of the manuscript that addresses the points raised during the review process.

We look forward to receiving your revised manuscript.

Kind regards,

Alejandro Huertas Herrera

Academic Editor

PLOS ONE

Journal Requirements:

3. Thank you for stating the following financial disclosure: "No external funding was used in the analysis represented by this manuscript, rather the United States Forest Service providing salary to support the involvement of all authors."

4. In the online submission form, you indicated that "The data underlying the results presented in the study are available from Michael Moore mooremj@iastate.edu."

6. Please upload a copy of Supporting Information Figure/Table/etc. S1 Table and S2 Table which you refer to in your text on page 54.

Additional Editor Comments:

Dear Authors,

After reviewing the manuscript, I believe it merits publication in PLOS ONE following revisions based on the reviewer`s comments.

Title:

• Better the short title: “Impacts of Forest Harvest Legacies on Stream Habitat in the Tongass National Forest, Alaska”

Materials and Methods:

• Include information about the restored and reference sites and provide details about the forests. Enhance the paper with relevant photos or illustrations that immerse the reader in the context of the forests. Additionally, incorporate more visual material related to the harvesting treatments, such as photos and illustrations.

• Provide additional information about the stream`s characteristics and consider including a map displaying the streams or an illustrative example.

• Expand on details about the floodplain.

• Enhance the information on the climate dataset, for example, specifying details like pixel size and the GIS geoprocessing involved.

• Relocate tables 1 and 3 to the appendixes.

Statistical Analyses:

• Eliminate subtitles.

• Clearly explain or provide more precise indications of the statistical analyses underlying Figures 4, 5, and 6. Consider moving relevant information from table and figure captions to the statistical analysis section.

Results:

• Move Table 4 to the appendixes.

Discussion:

• Condense the “Management Acting Upon the Template” section to a concise summary (no more than four paragraphs) that succinctly conveys the key points and transitions smoothly to the main ideas.

• Include a section discussing the weaknesses of the study.

• Consider adding a conclusion section for a more comprehensive overview.

Reviewers' comments:

Reviewer's Responses to Questions

**Comments to the Author**

1. Is the manuscript technically sound, and do the data support the conclusions?

Reviewer #1: Yes

2. Has the statistical analysis been performed appropriately and rigorously? 

Reviewer #1: Yes

3. Have the authors made all data underlying the findings in their manuscript fully available?

Reviewer #1: Yes

4. Is the manuscript presented in an intelligible fashion and written in standard English?

Reviewer #1: Yes

5. Review Comments to the Author

Reviewer #1: Dear author,

The manuscript entitled "Same streams in a diff­­erent forest? Investigations of forest harvest legacies and future trajectories across 30 years of stream habitat monitoring on the Tongass National Forest, Alaska" is a very interesting paper that captures the reader's attention. The paper is well written, and I think it is pleasant to read. This research emphasizes the imperative for sustainable practices to preserve fish habitats and maintain the overall health of southeast Alaska´s forest ecosystems. In my opinion, understanding these dynamics is crucial for anticipating and addressing future management challenges in this ecologically sensitive region. I enclose my specific comments in the PDF.

Bests

6. PLOS authors have the option to publish the peer review history of their article (what does this mean?). If published, this will include your full peer review and any attached files.

Reviewer #1: No

---

## [Author Response · Author response to Decision Letter 0]

19 Mar 2024

Response to Comments on Journal Requirements:

Note that all of this information is in an easier to read formatted word document version that was uploaded with the resubmission.

We changed figure and supplemental figure format to match in-text names as described in the link above.

Code has been cleaned and prepared for release. It will be released through GitHub upon publication of the work.

3. Thank you for stating the following financial disclosure: "No external funding was used in the analysis represented by this manuscript, rather the United States Forest Service providing salary to support the involvement of all authors."

We were a bit confused by the requests for additional details on the topic of funding. See our best attempts to address them below.

The source of funding was the United States Forest Service. No external funding was used in this study. 

No external funders had any role in study design, data collection and analysis, decision to publish, or preparation of the manuscript.

All authors received salary through the US Forest Service.

NA

Included in cover letter.

4. In the online submission form, you indicated that "The data underlying the results presented in the study are available from Michael Moore mooremj@iastate.edu."

Michael Moore is now a USGS employee. To avoid conflicting data sharing rules in the USFS and USGS, we would like to establish Rebecca Flitcroft rebecca.flitcroft@usda.gov as the primary contact for data and code for this research. The master dataset for this project is currently being reviewed for release through the US Forest Service’s repository, which will provide public access. Data will be available in repository upon publication.

6. Please upload a copy of Supporting Information Figure/Table/etc. S1 Table and S2 Table which you refer to in your text on page 54.

GIS Data used in figure 1 comes primarily from shapefiles we built from our own data from our analysis dataset that was collected by the USFS (now provided as supplementary info). The Alaska State Boundary Shapefile is owned by the United States Census and available for free use. The only other externally accessed GIS data would be the ArcGIS Pro World Ocean basemap. Basemaps can be used freely in academic publications as explained in ESRI online terms of use resources such as:https://doc.arcgis.com/en/arcgis-online/reference/static-maps.htm. and https://support.esri.com/en-us/knowledge-base/what-is-the-correct-way-to-cite-an-arcgis-online-basema-000012040, and through consultation with our USFS GIS specialist. In accordance with online guidelines for attribution and examples in other PlosOne publications, we added the following to the caption of Figure 1: 

“Alaska state boundaries shapefile produced by the United States Census Bureau was accessed from: https://www.census.gov/geographies/mapping-files/time-series/geo/carto-boundary-file.html. ESRI World Oceans Basemap accessed from: https://www.arcgis.com/home/item.html?id=67ab7f7c535c4687b6518e6d2343e8a2;

Sources: Esri, GEBCO, NOAA, National Geographic, DeLorme, HERE, Geonames.org, and other contributors,” on lines 63-87 of the revised. 

We reviewed references and made a few minor changes to citation styles.

Editor Specific Comments

Title:

• Better the short title: “Impacts of Forest Harvest Legacies on Stream Habitat in the Tongass National Forest, Alaska”

Change made on lines 5-6 of the revised.

Materials and Methods:

• Include information about the restored and reference sites and provide details about the forests. Enhance the paper with relevant photos or illustrations that immerse the reader in the context of the forests. Additionally, incorporate more visual material related to the harvesting treatments, such as photos and illustrations.

We agree that a visual illustration of stream sites in different management categories would be beneficial. Therefore, we created a three paneled map with a representative example of each treatment. We included this as the new S1 fig for now and adjusted other SFig numbers, but are open to its inclusion in the main text if the editors think it would be better placed there. We cite it in text on line 289 of the revised and add a caption as well that reads: “S1 Fig. Examples of stream sites in the “reference”, “harvested”, and “restored” management categories. Panel A shows a stream in reference condition with no riparian buffer and limited subwatershed harvest. Panel B shows a site with substantial buffer and subwatershed harvest. Panel C shows a restored site that had buffer harvest and wood was installed to help improve stream habitat conditions. Collectively, these panels illustrate the multi-storied forest condition of reference stands, the conversion to Alder dominated riparian forest post-harvest that is discussed in the paper, and it shows some installed wood pieces at restoration sites.”

• Provide additional information about the stream`s characteristics and consider including a map displaying the streams or an illustrative example.

We left stream lines off of the figure 1 map because the stream density in the stream network is too dense to show any meaningful information. See comment and figure example in response to comment on line 175 below.

We add additional information describing the characteristics of the primary process groups and provide citations of Paustian’s classification framework and the Tongass Management plan where these are described in greater detail on lines 241-253 of the revised. The added text is:

“Floodplain systems are low-gradient alluvial depositional channels, situated in valley bottoms and lowlands with high stream flows not commonly contained within the active channel banks and some degree of flood plain development is evident. The channels are predominantly composed of series of pools and riffles, with large wood the predominant pool forming mechanism. They often have more multi-threaded channels, greater sinuosity, and greater amounts of off channel habitats such as beaver ponds and sloughs [33]. The moderate gradient channels usually have channel gradients of 2-6 percent and occur near the transition between headwater streams and floodplain and alluvial fan channels. These streams usually have more confined valleys and coarser alluvial substrates ranging from gravel to boulder size comprising the channel beds and banks. Large woody debris can form log-step pools and lateral scour pools in these channels [33]. Across process groups a variety of stream channel sizes are represented ranging from channel bed widths of 1.9 to 50.1 meters.”

We also provide representative photos in Figure S1.

• Expand on details about the floodplain.

See previous comment where floodplain and stream channels characteristics were described in greater detail. 

• Enhance the information on the climate dataset, for example, specifying details like pixel size and the GIS geoprocessing involved.

The hydrologic classifications were developed by Sergeant et al. 2020. They classified stream flows from a complex runoff model for the Gulf of Alaska watershed that is based on a digital elevation model, landcover dataset, glacier inventory, and soil characteristics. We believe it is appropriate to add this short descriptor of the model and refer the reader to the source paper for a comprehensive presentation of the methodological details. The paper does not present the resolution of a raster model and summarizes the results to watersheds.

The revised sentence reads: “To capture differences in stream habitat that could be related to hydrologic regimes, we used published hydrologic classifications of rain-dominated and snow-dominated developed for watersheds in the Gulf of Alaska based on a classification of stream flows from a complex runoff model that incorporates a digital elevation model, landcover dataset, glacier inventory, and soil characteristics” On lines 278-285 of the revised. 

• Relocate tables 1 and 3 to the appendixes

Made change as suggested and renumbered main-text and supplemental figures and tables 

Statistical Analyses:

• Eliminate subtitles.

Deleted subtitles for statistical analyses. 

• Clearly explain or provide more precise indications of the statistical analyses underlying Figures 4, 5, and 6. Consider moving relevant information from table and figure captions to the statistical analysis section.

We added text explaining analyses from the figure captions to statistical analysis sections as explained below. We do not believe that this information should be removed from these figure captions because figure captions should be able to stand alone with the reader being able to interpret them without consulting the methods section of the paper. Our USGS Bureau approving official requires this level of detail in figure captions.

Additional explanation of marginal effects calculations that are shown in figure 4: “We calculated marginal effect sizes with our top models in order to illustrate the magnitude of expected changes in habitat metrics in response to predictor variable changes. To do this we predicted the percent change in the habitat metric response variable to a 25% change of the observed range of a given continuous management or intrinsic predictor variable while holding all other continuous predictor variables constant at their mean. For binomial predictors such as restoration status or hydroclass, we predicted a percent change in response variables from one level to the other.” on lines 364-370 of the revised.

Note that Figure 4 illustrates the model results for a single response variable. We feel that it is more appropriate to explain the results and methods in the figure caption that directly corresponds to that particular variable than to explain a single model in the methods section. 

For explanation of the plots in figure 5 representing

---

## [Editor Report · Decision Letter 1]

22 Mar 2024

Same streams in a diff­­erent forest? Investigations of forest harvest legacies and future trajectories across 30 years of stream habitat monitoring on the Tongass National Forest, Alaska

PONE-D-23-40089R1

Dear Dr. Moore,

We’re pleased to inform you that your manuscript has been judged scientifically suitable for publication and will be formally accepted for publication once it meets all outstanding technical requirements.

Kind regards,

Alejandro Huertas Herrera

Academic Editor

PLOS ONE

Additional Editor Comments (optional):

Dear Michael,

Thank you for addressing the revision suggestions for your manuscript and for your efforts in clarifying the issue regarding the data and code. Having thoroughly reviewed the new version of the manuscript, cover letter, response to the review file, and even the Author Query to Editor PONE-D-23-40089R1, I am confident that the paper merits publication in PLOS ONE.

Please consider the following minor details:

In both the manuscript and supporting information files, please add an asterisk next to Rebecca Flitcroft's name (Corresponding author): Flitcroft, R. L. 2*.

In Fig 2. (line 320), please include the abbreviation "AIC" (Akaike Information Criterion).

In the supporting information captions section, please ensure the following corrections:

Line 1105: Add "CBW - Channel Bed Width" to the main text.

Line 1110: Include "See table S1 and Table 1 for variable abbreviation explanations."

Line 1112: Add "Multi Response Permutation Procedure (MRPP)."

Please ensure consistency in caption information across both files.

Best regards,

Alejandro
---

## [Editor Report · Acceptance letter]

30 May 2024

PONE-D-23-40089R1 

PLOS ONE

Dear Dr. Moore, 

I'm pleased to inform you that your manuscript has been deemed suitable for publication in PLOS ONE. Congratulations! Your manuscript is now being handed over to our production team.

Kind regards, 

on behalf of

Dr. Alejandro Huertas Herrera 

Academic Editor

PLOS ONE